# Long-Term Vocational Outcome at 15 Years from Severe Traumatic and Non-Traumatic Brain Injury in Pediatric Age

**DOI:** 10.3390/brainsci13071000

**Published:** 2023-06-28

**Authors:** Sandra Strazzer, Valentina Pastore, Susanna Frigerio, Katia Colombo, Sara Galbiati, Federica Locatelli, Susanna Galbiati

**Affiliations:** Acquired Brain Injury Unit, Scientific Institute IRCCS E. Medea, 22040 Bosisio Parini, Italy; valentinapastore1@virgilio.it (V.P.); susanna.frigerio@lanostrafamiglia.it (S.F.); katia.colombo@lanostrafamiglia.it (K.C.); sara.galbiati@lanostrafamiglia.it (S.G.); federica.locatelli@lanostrafamiglia.it (F.L.); susanna.galbiati@lanostrafamiglia.it (S.G.)

**Keywords:** brain injuries, disorders of consciousness, childhood, follow-up studies, vocational outcome

## Abstract

Background: Recent studies suggest that acquired brain injury with impaired consciousness in infancy is related to more severe and persistent effects and may have a cumulative effect on ongoing development. In this work, we aim to describe vocational outcome in a group of patients at 15 years from a severe brain lesion they suffered in developmental age. Methods: This study included a total of 147 patients aged 1.5 to 14 years with acquired brain lesion. Clinical and functional details (“Glasgow Outcome Scale”, “Functional Independent Measure” and Intelligence Quotient) were collected at the time of their first hospitalization and vocational outcome was determined after 15 years. Results: 94 patients (63.9%) presented with traumatic brain injury, while 53 patients (36.1%) presented with a brain lesion of other origin. Traumatic patients had a higher probability of being partly or fully productive than non-traumatic ones: 75.5% of traumatic subjects were working—taking into account limitations due to the traumatic event—versus 62.3% of non-traumatic ones. A relationship between some clinical variables and the vocational outcome was found. Conclusions: Rehabilitation should adequately emphasize “vocational rehabilitation” because a significant proportion of people experiencing a disorder of consciousness in childhood may show good social integration in adult age.

## 1. Introduction

Severe acquired brain injury (ABI) causing transient or permanent disorders of consciousness (DoC) in pediatric age is a rare but life-long impacting condition, with increasing epidemiology in Europe and the rest of the world [1,2].

Advances in resuscitation practice are gradually increasing survival rates for DoC patients. Furthermore, global healthcare improvements in many Low- and Middle-Income Countries (LMICs) are widening the geography of survival. This results in more patients requiring long-term care, as well as specialized and customized rehabilitation [3,4,5].

The natural history and prognosis of children with DoC are not well-defined [6]; therefore, accurately assessing the presence or absence of a particular DoC and understanding typical recovery trajectories serve as prerequisites for defining appropriate treatment goals and providing support and counseling to families for adequate expectations [7,8,9,10]. Sequelae in children and adolescents after severe ABI frequently include motor, cognitive and language impairments [11,12,13]. Several studies also suggest that moderate-to-severe ABI children and adolescents often also show relevant behavioral problems, psychological impairments, low emotional regulation and difficulties in adaptive functioning [14,15,16,17]. It is now recognized that these difficulties have an even more dramatic impact on young patients than what is commonly observed in adults, mainly because of the greater vulnerability of a yet immature brain and the potential of an early ABI to permanently alter and change the complex processes of cognitive development [18,19].

On the whole, these difficulties have the potential to make transition from childhood and adolescence to adult age particularly stressful and complicated [20]. In addition, another crucial challenge for young patients with ABI upon reaching adult age is occupational life. In our culture, employment is crucial in determining self-esteem and symbolizes full integration into, and belonging to, a community [21]. For this reason, successful medical rehabilitation may end in failure if occupational integration in adult age becomes a stumbling block for patients suffering a brain lesion in pediatric age, [22], with a number of impactful consequences at a psychosocial and economic level [23]. Several studies have reported on brain lesion outcomes up to 10 years post-injury [24,25], but the scientific literature about outcomes up to 20 years is scant [26,27,28,29,30], and few studies have been conducted in Europe [31,32,33]. Additionally, while this holds true for adult patients suffering an ABI, it is even more so for patients who suffered a brain insult early in life. As regards adults, most studies suggest a strong relationship between disability and occupational status, and the community integration levels are considerably below those reported for non-disabled patients, forcefully impacting on the perceived quality of life. Except for a study published in 1993 on children with head injuries varying in severity [34], there is no corresponding literature. However, we believe that the earlier the brain damage, the more dramatic the possible impact of a severe brain lesion on vocational outcome as the main developmental stages would be impaired and relational experiences would be limited with consequences on social skills and individual autonomy. In other words, it would affect all those abilities underlying a satisfactory working life in adult age.

For these reasons, the present study aims:-To describe vocational outcome in a group of ABI individuals 15 years after they suffered a brain lesion in developmental age;-To determine the association with variables related to time of injury, demographic variables and clinical picture;-To compare vocational outcome in patients with Traumatic Brain Injury (TBI) vs. a sample of patients with ABI of other origin (anoxic, infective or vascular);-To identify patients presenting a more marked risk for difficult social and occupational integration.

## 2. Materials and Methods

### 2.1. Participant

This study recruited patients from a cohort of children with ABI who were admitted to the “Eugenio Medea” Scientific Institute for a clinical functional assessment and for rehabilitation interventions during the 1999–2004 period and a later follow-up in 2015–2019.

Inclusion criteria for Group 1 (patients with TBI) and Group 2 (patients with a ABI of other etiology) were the same: (i) age at insult <14 years; (ii) documented evidence of severe ABI (Glasgow Come Scale (GCS) < 9) [35]; (iii) medical records sufficiently detailed to determine injury severity and neurological findings; (iv) no documented history of neurological or developmental disorders (i.e., autism, learning or attention disorders) or previous ABI; and (v) no pre-existing acute or chronic serious illnesses.

### 2.2. Measures

The following clinical and medical data were collected for each patient: sex, age at diagnosis, age at the time of assessment, days of unconsciousness, neurosurgery, site of brain lesion, drug therapy at assessment and presence of motor, visual and language problems.

The “Glasgow Outcome Scale” (GOS) [36] score, the “Functional Independent Measure for Children” (WeeFIM) [37] and the “Functional Independent Measure” (FIM) [38] scores and the Intelligence Quotients were also collected.

The GOS is used to assess outcome after a severe brain injury and is based on five global categories (Death, Persistent Vegetative State, Severe Disability, Moderate Disability and Good Recovery).

The Wee-FIM and FIM measure overall functional disability in children and adults, respectively, by 18 items assessing three domains (self-care, mobility and cognition). Each item can be scored from 1 (complete dependence/total assistance) to 7 (complete independence/no helper); total scores range from 18 to 126. Scores reported in our study were normed to 100 based on age.

We also collected the Intelligence Quotients measured upon the first admission and administered the Wechsler Adult Intelligence Scale—Revised (WAIS-R) [39] at the 15-year follow-up. The children’s familial socioeconomic status (SES) was described with the “Four Factor Index of Social Status” by A.B. Hollingshead [40], one of the most widely used socioeconomic classifications. According to this index, social status is a multidimensional concept, including at least four variables: education, occupation, sex and marital status.

### 2.3. Procedures

The study protocol was approved by the local research Ethics Committee and all participants (or their caregivers) gave their written informed consent in line with the Declaration of Helsinki.

None of the patients who had been asked to participate in this study refused to take part.

At the time of first hospitalization, clinical and demographic details were obtained from medical records and from clinical interviews with caregivers during the subacute phase. After fifteen years, details about the participants’ vocational outcome and their current general clinical picture were obtained during a follow-up evaluation, if possible. When this was not possible, a telephone interview was planned.

### 2.4. Statistical Analysis

Data are presented as means, standard deviations (SDs) and percentages. The TBI group and the comparison group were compared on demographic and clinical factors using Chi-square statistics and t-tests, as appropriate, to study differences between groups.

Hierarchical regression analysis was used to determine predictors of three variables (GCS score, age at injury and SES) on vocational outcome as the dependent variable. For both groups, variables were individually entered: GCS score was entered at step 1, age at injury was entered at step 2 and SES at Step 3.

Statistical analysis was performed using SPSS Statistics 24 (SPSS.24, Inc., Chicago, IL, USA). Significance was set at a *p* value of <0.05.

## 3. Results

### 3.1. Upon Admission

In the 1999–2004 period, a total of 220 patients aged 1.5 years to 14 years with a diagnosis of acquired brain lesion of traumatic, anoxic, vascular, infectious or oncological etiology were admitted to our Scientific Institute, where they received a clinical functional assessment and rehabilitation.

A total of 12 patients (3 patients with TBI and 9 patients with brain lesions of other origin) died during the study period, with a mean temporal distance from insult of 8.2 years (SD = 4.4 years) for patients with TBI and of 5.0 years (SD = 3.5 years) for patients with lesions of other origin; 9 patients could not be included because of a previous diagnosis; 52 patients dropped out at the 15-year follow-up (in most cases, they were not traceable because of a plausible change in residence between first hospitalization and follow-up).

A total of 147 patients (102 males and 45 females) met the inclusion criteria and were therefore included in the study. Group 1 (patients with TBI) included 94 patients (63.9% of the total sample), whereas Group 2 (patients with brain lesions of other origin) included 53 patients (36.1%) (Figure 1). In more detail, Group 2 included 17 patients with acquired brain lesion of anoxic etiology (11.6%), 20 patients with lesions of vascular etiology (13.6%), 9 patients with lesion of infectious etiology (6.1%) and 7 patients with lesions of oncological etiology (4.8%).

Table 1 describes the clinical and demographic characteristics of the total sample and of Group 1 and Group 2, separately.

### 3.2. Group 1: Patients with TBI

Mean age at traumatic injury was 8.55 years (SD = 3.91) with a mean time between insult and hospitalization slightly longer than 1 year. The mean GCS score (X = 5.33, SD = 1.71) was indicative of severe TBI. Mean period of unresponsiveness was remarkable: 32.26 days (SD = 38.96). On the whole, patients showed a condition characterized by a moderate–severe disability as a consequence of brain lesion, as shown by their mean GOS score (X = 3.16, SD = 1.05). Additionally, the Wee-FIM scores suggested a moderate level of assistance (X = 46.56, SD = 36.55). Subjects also showed a borderline cognitive level (X = 72.85, SD = 19.33). Their mean SES score was 36.83 (SD = 12.38), corresponding to a quite good standard of living. A significant percentage of subjects with TBI showed motor, visual or behavioral problems. Most subjects did not present seizures but at the time of their first evaluation. A total of 76.6% of the sample was on drug therapy (antiepileptics were the most frequently administered drugs, followed by antacids and gastric protectors). From a neurological point of view, almost all the subjects with TBI presented with diffuse brain damage involving at least one brain hemisphere. More specifically, the neuroradiological examination revealed that most patients reported multifocal/diffuse lesions (57.4% of the sample).

### 3.3. Group 2: Patients with Brain Lesions of Other Origin

Group 2 presented with sociodemographic and clinical characteristics comparable to those described for Group 1, with the only exception of the following three clinical variables: epilepsy, drug therapy and site of lesions. More precisely, patients from this group had a higher probability of presenting with seizures (χ^2^ = 4.081; *p* = 0.043), received drugs in a statistically significant higher percentage (χ^2^ = 5.853; *p* = 0.000) and showed a higher probability of focal brain lesions (χ^2^ = 30.565; *p* = 0.000)

### 3.4. Follow-up at 15 Years

In the 2015–2019 period, patients from both groups were assessed again by a clinician directly involved in this study. Depending on the specific situation of each subject, the assessment took place directly in our Institute (*n* = 96, 65.3%) or by phone (*n* = 51, 34.7%). In general, telephone interviews were more frequent with severe clinical pictures. When patients could not respond by themselves, their main caregivers were involved.

GOS, FIM and the IQ scales were administered again. Vocational outcome was also determined and classified according to the three groups below:

Not productive: patients with a very severe/severe clinical picture who receive a disability allowance or patients with a less severe picture who are involved in any working activity.

Partly productive: patients with a moderately severe picture, with or without a disability allowance. They can either attend a daily center or, in some cases, have a part-time paid job. Patients attending, for example, university or training courses, are included in this group.

Fully productive: patients with a competitive, full-time paid job.

### 3.5. Group 1: Patients with TBI

At the 15-year follow-up, patients with TBI presented with a mean GOS score of 3.70 (SD = 0.80). Despite being slightly higher than the initial admission score, it indicates a moderate–severe disability. Mean FIM score was 92.58 (SD = 36.26), indicating a lower level of assistance than at the first evaluation. Finally, the mean IQ score (X = 76.40, SD = 23.89) was slightly higher than after insult (but 15 patients could not be evaluated). Looking at the vocational status, as reported in Table 2, 23 patients (24.5%) were not productive, 41 patients (43.6%) were partially productive and 30 patients (31.9%) were fully productive. More precisely, 11 patients of our sample (11.7%) presented a very severe outcome (Vegetative State or Minimally Conscious State), and therefore they lived at home with caregivers and could not have a working life; 28 patients (29.8%) attended a daily center and in some cases had a part-time job; 30 patients (31.9%) had a full-time work activity; and 13 subjects (13.8%) were involved in a study activity. Finally, 12 patients did not work or study despite a good outcome and the potential opportunity to work.

### 3.6. Group 2: Patients with Brain Lesions of Other Origin

Fifteen years after their first hospitalization, patients in this group also presented with a moderate–severe level of disability assessed by GOS (X = 3.52, SD = 0.91). Their level of independence as assessed by FIM (X = 80.49, SD = 44.69) was increased as compared to their first assessment. Their mean IQ scores remained in the borderline range but it was slightly higher (X = 75.17, SD = 23.41; for this variable, 12 patients were not evaluable).

As observed at the first hospitalization evaluation and at the follow-up, too, no statistically significant differences were found for these three functional variables (GOS, FIM, IQ) in patients with TBI vs. patients with brain lesions of other origin.

Looking at vocational outcome, 20 patients (37.7% of the total group) were not productive, 20 patients (37.7%) were partially productive and 13 patients (24.5%) were fully productive. In more detail, 13 patients (24.5%) were not productive because of a clinical condition characterized by a severe disability, 10 patients (18.9%) attended a daily center and in some cases had a part-time job, 13 patients (24.5%) had a full-time paid job, 10 patients (18.9%) attended a high school or a university and 7 patients (13.2%) did not study or work, although their condition was not severe.

### 3.7. Regression Analyses

A hierarchical regression analysis was carried out for both groups in relation to three relevant clinical variables: GCS score at insult, age at injury and socioeconomic status. Vocational outcome (categorized as “not productive, “partly productive” and “fully productive”) was used as the dependent variable. See Table 3 for the significant relationship we found.

### 3.8. Group 1: Patients with TBI

For patients suffering a TBI, two variables were predictive of vocational outcome: the GCS score (β = 0.206; *p* = 0.046) and the socioeconomic status according to the family SES score at the first hospitalization (β = 0.261; *p* = 0.012). Both the GCS score and the SES score were higher for patients who were fully productive, whereas for not productive or partly productive patients, these mean values were substantially similar. On the contrary, age at injury was not predictive of working activity.

### 3.9. Group 2: Patients with Brain Lesions of Other Origin

None of the three clinical variables considered were predictive of vocational outcome at follow-up.

## 4. Discussion

A satisfactory integration into working life in adult age is a vital goal in the rehabilitation of children and adolescents suffering a severe ABI early in life, as it supports and protects their psychological and social well-being [30].

The scientific literature mainly focuses on adult patients [41,42,43,44], while the impact of brain damage sustained in pediatric age on the future occupational life of patients has been insufficiently studied [35]. Furthermore, the literature rarely describes the very-long-term outcome, even if this is an important aspect to keep track of the natural history of the pathology and enables a more accurate prognosis and more appropriate rehabilitation programs, with beneficial effects for patients and their families [6].

Here, we focused on the long-term vocational outcome in a wide sample of young adults who suffered an ABI during pediatric age, and compared TBI vs. lesions of non-traumatic etiology. A relationship between the vocational outcome and some relevant clinical and sociodemographic variables emerged.

Our sample of patients suffered a brain lesion at a very early age and during school age (their mean age was slightly higher than 8 years). All patients sustained severe brain damage, as confirmed by the mean length of the period of unresponsiveness (about one month). At the first hospitalization, their mean level of disability was moderate–severe and they generally required a moderate level of assistance from their caregivers. Most patients had a borderline cognitive profile; most families of children and adolescents involved in this study had a good standard of living. Motor, sensorial and behavioral problems were particularly common in both groups of patients [45]. Multifocal lesions were prevalent for patients with a traumatic brain injury, while focal lesions were dominant for patients with lesions of different origin. This was, however, predictable given the same etiology of the brain damage.

At the 15-year follow-up, patients from both groups still presented with a moderate–severe disability, although they required a slightly lower level of assistance. Their mean cognitive profile remained in the borderline range.

The most interesting piece of information was related to their occupational status.

Patients with ABI of traumatic etiology had a higher probability of being partly or fully productive than patients with brain lesions of other origin. Nearly half of the patients with TBI were partly productive, whereas almost one third of them had a full-time activity. Therefore, on the whole, 75.5% had an occupation in line with their physical and cognitive limitations. Twenty-three patients with TBI were not productive at all: some of them were too impaired to engage in any form of professional activity, while twelve patients had a globally good outcome but were inactive.

Considering patients with a brain damage of vascular, anoxic or infective etiology, 62.3% were partly or fully productive. As regards the category “not productive” and contrary to patients with TBI, the proportion of patients that could not work because of a severe outcome was higher (13 patients out of a total number of 20), while 7 patients could work but did not.

Although, as previously underlined, no significant differences emerged between the two groups as regards the clinical and demographic characteristics at the first hospitalization, we could suppose that the global impairment of patients with brain damage of other origin—although just slightly higher than that of patients with traumatic brain lesion—can justify this. However, the number of non-working patients with a non-severe outcome is in general not negligible, and it would be interesting in a future study to better explore the reason why subjects with a good outcome show this difficulty. The limited social skills of patients who survived a brain lesion may possibly play a crucial negative role.

The regression analysis suggested different considerations for the two groups of this study. These considerations should, however, be regarded with due caution given the limited sample size. Further evidence from larger-scale studies is needed. In any case, we considered the impact of three relevant clinical variables (GCS score at insult, age at injury and familial socioeconomic status of patients) on the vocational outcome 15 years after insult.

None of these variables were predictive for patients with brain lesions of other origin. We could not compare this with previous studies as we could not find any on this topic, on this category of patients and, above all, on this age group.

By contrast, for patients with TBI, two variables were predictive of vocational outcome: GCS score and socioeconomic status. Both were higher for fully productive patients, while the two mean values were substantially similar for non-productive patients and partly productive patients.

As regards the first variable, the scientific literature focusing on patients suffering a TBI in adult age suggests that injury severity and, therefore, the GCS score, have a strong association with return to productivity [46,47]. Of course, this has been reported for adult patients but can possibly be extended to patients in developmental age.

Familial socioeconomic status had an association with occupational status, and in particular, patients belonging to families with a higher socioeconomic condition had a higher opportunity to have a full-time job. Previous studies on adults [48,49] confirmed this trend, although in this case the most frequently considered indicator is the educational level of patients (on the contrary, SES is related to a multidimensional concept, including but not limited to education). Higher-SES families are expected to have more educational and economical resources available and can gain earlier access to specific rehabilitation during the different developmental stage of their children. Furthermore, they can most likely support their grown-up children in searching for a satisfactory job, taking into consideration their physical and cognitive limitations.

A short final consideration on age at injury—the other variable taken into account for regression analyses. This variable did not prove predictive for the vocational outcome for both groups. Several studies on adult samples [32,50] indicate that this variable is a strong predictor of return to productivity and of employment status: the younger a person was at the time of injury, the more likely they would return to productivity. Authors in general have considered this as a consequence of the greater efforts made to rehabilitate younger patients and also of the plasticity of their brain, which decreases over the years. In addition, another explanation is that younger subjects tend to adapt to new conditions with less effort or can even accept jobs that require low qualifications. In our sample, the particularly low mean age at injury and the reduced dispersion index for both groups (SD = 3.89 years for total sample) could make the impact of this variable less crucial in our statistical analysis.

## 5. Conclusions

The natural history and prognosis of children with DoC are not well-defined [6]. To our knowledge, this is the first study to describe the long-term vocational outcome of patients suffering a severe ABI of different origin in pediatric age.

The findings of this study must be considered in the context of the following limitations. First, Group 2 includes subjects with heterogeneous diagnosis. Future studies should differentiate and compare subjects with vascular, anoxic or infective brain lesions. Second, we could not describe in detail the behavioral problems of patients (i.e., by use of structured instruments such as the SCID-I or SCID-II) [51], because this decisive information was not collected. This piece of information should be included in future research, mainly in consideration of the impact of psychiatric and behavioral symptoms on the working life of patients with acquired brain lesions.

Despite these limitations, we believe that our results can be used in clinical practice to predict long-term occupational outcome for patients who suffered a severe ABI in pediatric age. While patients with mild or moderate brain lesions can in most cases recover rapidly and can thus aim for a gratifying adult life, a large proportion of severely injured patients can live with permanent impairments profoundly affecting their personal and human potential. Given the crucial impact of professional and occupational satisfaction on our lives, we strongly believe that medical rehabilitation should also adequately and early emphasize “vocational rehabilitation” [23], which is even more important when the brain lesion occurs during childhood and adolescence. Future research on this topic should focus on adequately supporting rehabilitation programs for young patients in this domain. We strongly believe that such important insights could also guide national policies.

## Figures and Tables

**Figure 1 brainsci-13-01000-f001:**
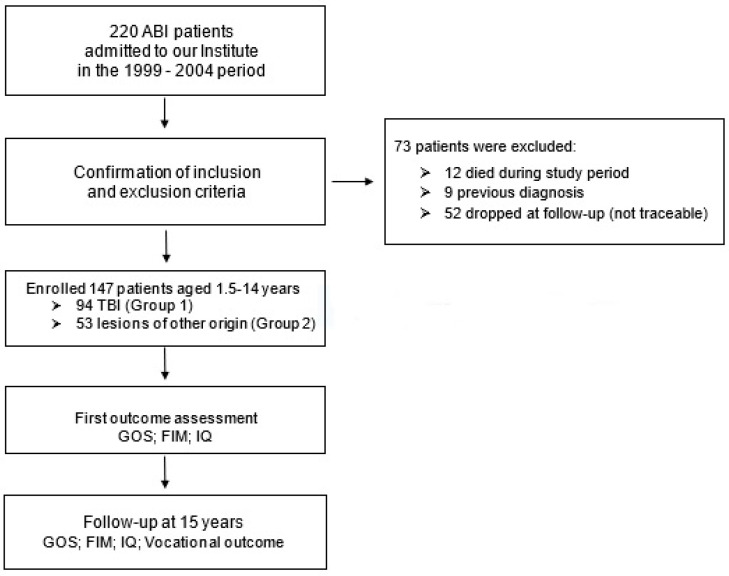
Flowchart outlining the process over time.

**Table 1 brainsci-13-01000-t001:** Clinical and demographic characteristics of the clinical sample and for the two etiology groups separately.

	Total Sample (*n* = 147)		Group 1 Traumatic Brain Injury (*n* = 94)		Group 2 Other Brain Lesion (*n* = 53)		
	Mean	SD	Mean	SD	Mean	SD	T (*p*)
Age at pathological event (years)	8.76	3.89	8.55	3.91	9.11	4.03	Ns
Days of unconsciousness	30.28	38.26	32.26	38.96	25.83	36.90	Ns
GCS	5.48	1.78	5.33	1.71	5.74	1.88	Ns
GOS	3.08	0.96	3.16	1.05	2.94	0.77	Ns
Wee-FIM	47.03	35.19	46.56	36.55	47.78	33.33	Ns
SES	36.51	12.08	36.83	12.38	35.94	11.62	Ns
Full IQ *	72.29	20.23	72.85	19.33	71.12	22.28	Ns
Sex							Ns
Females	45	30.6	24	25.5	21	39.6	
Males	102	69.4	70	74.5	32	60.4	
Neurosurgery							Ns
No	79	53.7	51	54.3	28	52.8	
Yes	68	46.3	43	45.7	25	47.2	
Motor problems							Ns
No	24	16.3	17	18.1	7	13.2	
Hemiparesis	41	27.9	25	26.6	16	30.2	
Tetraparesis	54	36.7	32	34.0	22	41.5	
Ataxia	9	6.1	6	6.4	3	5.7	
Motor control problems	13	8.8	9	9.6	4	7.5	
Paraparesis	4	2.7	4	4.3	0	0	
Visual problems							Ns
No	82	55.8	52	55.3	30	56.6	
Hemianopsia	6	4.1	6	6.4	0	0	
Blindness (<2/10)	9	6.1	3	3.2	6	11.3	
Deficit (<7/10)	24	16.3	15	16.0	9	17.0	
Not evaluable	26	17.7	18	19.1	8	15.1	
Behavioral problems							Ns
No	49	33.3	31	33.0	18	34.0	
Yes	76	51.7	50	53.2	26	49.1	
Not valuable	22	15.0	13	13.8	9	17.0	
Epilepsy							χ^2^ = 4.081; *p* = 0.043
No	123	83.7	83	88.3	40	75.5	
Yes	24	16.3	11	11.7	13	24.5	
Drug therapy							χ^2^ = 5.853; *p* = 0.016
No	26	17.7	22	23.4	4	7.5	
Yes	121	82.3	72	76.6	49	92.5	
Site of the lesion							χ^2^ = 30.565; *p* = 0.000
Multifocal	74	50.3	54	57.4	20	37.7	
DAI	18	12.2	18	19.1	0	0	
Frontal focal	15	10.2	6	6.4	9	17.0	
Posterior focal	13	8.8	5	5.3	8	15.1	
Other	22	15.0	7	7.4	15	28.3	
No lesion	5	3.4	4	4.3	1	1.9	

* A total of 41 patients were not evaluable for IQ values. Therefore, this variable was considered for *n* = 106 (more precisely, patients with TBI *n* = 72, patients with other lesions *n* = 34). DAI is diffuse axonal injury.

**Table 2 brainsci-13-01000-t002:** Vocational outcome at follow-up for patients with TBI (Group 1) and patients with brain lesions of other origin (Group 2).

	*n (%*)
Not productive	
Patients with TBI (Group 1)	23 (24.5%)
Patients with brain lesion of other origin (Group 2)	20 (37.7%)
Total sample	43 (29.2%)
Partly productive	
Patients with TBI (Group 1)	41 (43.6%)
Patients with brain lesion of other origin (Group 2)	20 (37.7%)
Total sample	61 (41.6%)
Fully productive	
Patients with TBI (Group 1)	30 (31.9%)
Patients with brain lesion of other origin (Group 2)	13 (24.6%)
Total sample	43 (29.2%)

**Table 3 brainsci-13-01000-t003:** Results of regression analyses for the two groups with vocational outcome at follow-up as dependent variable.

Group	Beta Coefficient	*p*-Value
Patients with TBI (*n* = 94)		
GCS score	0.206	0.046
Age at injury	−0.035	ns
SES	0.261	0.012
Patients with brain lesion of other origin (*n* = 53)		
GCS score	−0.119	ns
Age at injury	−0.002	ns
SES	0.126	ns

## Data Availability

The data presented in this study are available on request from the corresponding author. The data are not publicly available due to privacy.

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
