# Peer review of "Long-Term Vocational Outcome at 15 Years from Severe Traumatic and Non-Traumatic Brain Injury in Pediatric Age"

_brainsci, 2023, doi:10.3390/brainsci13071000_

Round 1

Reviewer 1 Report

Comments and Suggestions for Authors

Dear authors, it is a very interesting, high-quality, and transparent piece of work. I would like to suggest two aspects that could enhance the readability. First, I recommend including a flowchart outlining the process over time, including the number of participants at the start, the instruments used, the number of patients lost during follow-up, and the reasons for their loss. Secondly, I suggest updating the references used in the discussion section, as it requires a higher percentage of articles from the past 5 years, ideally from the last year.

Reviewer 2 Report

Comments and Suggestions for Authors

The manuscript is written very concisely and the study is very relevant due to the long follow-up period. I see potential for improvement especially in the presentation of the results:

 -Line 177, p5 ("Group 2: ..."): Please edit this line as a subheading (i.e. italics and spacing)

- Line 192: Please edit the formatting of the three groups

- Table 2: Please add data for Group 2

- Table: It’s sufficient if "n (%)" is noted once above the figures - it does not have to appear separately in each row

- Please add case numbers for both groups to table 3.

- Please add a table with GPOS-E, WeeFIM and FIM scores of both measurement occasions for both groups (only including cases for which paired scores are available).

- Discussion section: Please indicate that significance testing within regression analysis is also confounded by sample size. This should be considered when discussing if and why a predictor is significant or not.

- Inverted commas in the "Author Contributions" section should be removed.

Comments on the Quality of English Language

---

Reviewer 3 Report

Comments and Suggestions for Authors

In this work, the authors were interested in the implications for professional life and social integration of patients with severe disorders of consciousness that occurred in childhood. In their clinical study, they contacted 147 patients or their caregivers for those most severely disabled (group 1, 94 with traumatic brain injury and group 2, 53 with anoxic, vascular or infection brain lesions) 15 years after their brain injury in pediatric age (1.5 to 14 years old). Clinical and demographic characteristics of the two groups were comparable except concerning epilepsy, drug therapy and focal brain lesions statistically more present in group 2. Vocational outcome at 15 years follow-up showed that 71 out of 94 (75%) traumatic brain injury patients were partly of fully productive. They were only 33 out of 53 (62%) in group 2. Age at injury was not predictive of vocational outcome. Glasgow Coma Scale and the children’s familial socioeconomic status were predictive of vocational outcome for group 1 only.

All these points are discussed, argued and criticized by the authors who thus contribute to improving the care and long-term follow-up of young people with disorders of consciousness.

Minor comments:

Line 60. Scientific literature about outcomes up to 20 years is scant and a bit old.

Line 112. In this chapter, you define the scale of familial socio-economic status of patients without formally introducing the acronym SES used in rest of your paper.

Line 159. DAI not defined in table 1. Please specify “diffuse axonal injury” in the table legend. The other acronyms are already defined in the manuscript.
